# Exponentially Long Transient Time to Synchronization of Coupled Chaotic Circle Maps in Dense Random Networks

**DOI:** 10.3390/e25070983

**Published:** 2023-06-27

**Authors:** Hans Muller Mendonca, Ralf Tönjes, Tiago Pereira

**Affiliations:** 1Instituto de Ciências Matemáticas e Computação, Universidade de São Paulo, São Carlos 13566-590, SP, Brazil; hans.mendonca@usp.br (H.M.M.); tiago@icmc.usp.br (T.P.); 2Institute of Physics and Astronomy, Potsdam University, 14476 Potsdam-Golm, Germany

**Keywords:** synchronization, random networks, chaotic maps, mean-field analysis, finite size effects

## Abstract

We study the transition to synchronization in large, dense networks of chaotic circle maps, where an exact solution of the mean-field dynamics in the infinite network and all-to-all coupling limit is known. In dense networks of finite size and link probability of smaller than one, the incoherent state is meta-stable for coupling strengths that are larger than the mean-field critical coupling. We observe chaotic transients with exponentially distributed escape times and study the scaling behavior of the mean time to synchronization.

## 1. Introduction

Complex nonlinear systems often exhibit collective synchronization phenomena which can play an important role for the overall functioning of a system [1,2,3]. Phase oscillator models can elucidate key aspects of the mechanism that generates the collective motion [4]. The Kuramoto model, for instance, is particularly useful in describing groups of weakly coupled oscillators such as Josephson junctions, and they can be analyzed in almost full detail in the thermodynamic limit of infinitely many oscillators. Indeed, Kuramoto himself initially studied the fully connected networks of coupled oscillators with frequency heterogeneity, and obtained the critical value of the coupling strength for the transition from incoherence to synchronized collective oscillations [5].

While such predictions are obtained in the thermodynamic limit, they have been used as fruitful approaches to describe networks with finitely many oscillators [6,7]. However, recent work has shown that finite size fluctuations or sparse connections in the network can significantly impact on the overall dynamics. In fact, in certain models, synchronization cannot, even approximately, be predicted from the mean-field approximation in the thermodynamic limit [8]. That is, in these models, a transition to synchronization occurs or is inhibited because of finite size fluctuations [9,10]. The interplay between mean-field predictions and finite-size fluctuations for general models remains elusive and requires further investigation.

In this work, we study chaotic phase maps in dense networks where the mean-field dynamics can be analyzed exactly in the thermodynamic limit. For small coupling, due to the chaotic phase dynamics, only incoherence is stable. For a range of coupling strengths, mean-field analysis predicts coexistence between complete chaotic synchronization and incoherence, and for strong coupling, the incoherence becomes unstable. Then, complete synchronization is the globally attracting state in our model. Our results are two-fold:

(i) For coupling strengths with a stable coexistence of incoherence and synchronization, although incoherence is locally attracting, finite-size fluctuations can take the system into the basin of attraction of the absorbing state of complete synchronization. Starting near incoherence with uniformly distributed random oscillator phases, the distribution of transient times towards synchronization is exponential and scales as a power of the system size.

(ii) Above the critical coupling strength, in dense but incomplete networks, although linear stability analysis of the mean-field equations suggests that any nonzero mean field, e.g., finite size fluctuations of the mean field, will grow exponentially fast, we observe an exponentially long chaotic transient in the incoherent state. Such a delayed transition to synchronization has so far not been described in dense networks of coupled phase oscillators or coupled chaotic maps.

## 2. Model of Coupled Chaotic Maps

The local phase dynamics in each node is modelled as a Bernoulli map of the circle with time steps t∈Z
(1)φ(t+1)=f(φ(t))=2φ(t)mod2π,
or via the abuse of notation on the complex unit circle z=exp(iφ), we write $z\left(t+1\right)=f\left(z\left(t\right)\right)=z{\left(t\right)}^{2}$. This map is chaotic and structurally stable [11]. That is, the statistical properties of the map persist under small perturbations. Therefore, for small coupling, the maps behave as nearly independent, and no collective dynamics is possible for small coupling. In [12], the global coupling of the phase dynamics is implemented as a Moebius map on the complex unit circle. The Moebius map has been shown to give exact solutions of sinusoidally forced phase dynamics [13], including the Kuramoto model, Winfree-type phase equations, and via a nonlinear transformation, the dynamics of theta neurons [14]. It is therefore a meaningful alternative to the sine coupling in the standard circle map. Here, we use a composition of (Equation 1) and a Moebius map (see Figure 1)
(2)$$z\left(t+1\right)=M\left(f(z(t)),\Phi (t),\tau (t)\right),$$
where
(3)$$M\left(w,\Phi ,\tau \right)=\frac{e^{i\Phi }\tau +w}{1+e^{-i\Phi }\tau w}$$
for a coupling intensity −1<τ<1, an angle of contraction $\Phi \in S^{1}$, and a point w∈D={z∈C:|z|<1} on the open complex unit disc. The family of Moebius maps is a group of biholomorphic automorphisms of D, and via analytic continuation, these transformations map the boundary of D bijectively onto itself. The effect of (Equation 3) on the unit disc is a contraction of almost all points towards exp(iΦ) on the boundary where $\lim_{\tau \to \pm 1}M\left(w,\Phi ,\tau \right)=\pm \exp\left(i\Phi \right)$ and $\lim_{\tau \to 0}M\left(w,\Phi ,\tau \right)=w$. The parameter τ characterizes the strength of the contraction. For τ→0, the map (Equation 2) approaches the uncoupled dynamics (Equation 1). Moreover, the family of wrapped Cauchy distributions
(4)$$p\left(\varphi \right)=\frac{1}{2\pi }\frac{1-R^{2}}{|1-Re^{i(\varphi -\Theta )}{|}^{2}}$$
which includes incoherence as the uniform distribution when R→0 and a delta distribution at φ=Θ when R→1, is invariant under (Equation 2) and (Equation 3) [12,13,15]. This family of continuous phase measures, in the context of phase synchronization, is known as the Ott-Antonsen manifold, and assuming this form of phase distribution is equivalent to the so called Ott-Antonsen ansatz [16,17]. The Ott-Antonsen manifold is parameterized using the mean-field amplitude *R* and the mean-field angle Θ
(5)$$Z=Re^{i\Theta }=\int_{0}^{2\pi }e^{i\varphi }p\left(\varphi \right)\;d\varphi .$$

The mean-field amplitude *R* is the Kuramoto order parameter [18], which is zero for incoherence, i.e., a uniform phase distribution, and R=1 for complete synchronization $\varphi_{n}=\Theta$ (a.s.). Furthermore, the higher circular moments $Z_{q}$ on the Ott-Antonsen manifold with q∈Z are integer powers of the mean field
(6)$$\begin{array}{c} Z_{q}=\int_{0}^{2\pi }e^{iq\varphi }p\left(\varphi \right)\;d\varphi =Z^{q}. \end{array}$$

As a consequence, phase doubling maps the circular moments as $f\left(Z_{q}\left(t\right)\right)=Z_{2q}\left(t\right)=Z_{2}^{q}\left(t\right)=f{\left(Z_{1}\left(t\right)\right)}^{q}$, leaving the Ott-Antonsen manifold invariant and mapping the mean-field amplitude and phase as $R\to R^{2}$ and Θ→2Θ.

To couple the dynamics of the Bernoulli maps (Equation 2), the parameters Φ(t) and τ(t) in (Equation 3) should be defined as functions of the ensemble mean field. Following [12], we define the contraction angle Φ(t) and the coupling intensity τ(t) as
(7)$$\begin{array}{ccc} Z(t) & = & \frac{1}{N}\sum_{n=1}^{N}z_{n}\left(t\right)=R\left(t\right)e^{i\Theta (t)} \end{array}$$
(8)$$\begin{array}{ccc} \Phi (t) & = & 2\Theta (t) \end{array}$$
(9)$$\begin{array}{ccc} \tau (t) & = & \tanh\left(\frac{\varepsilon }{2}R\left(t\right)\right), \end{array}$$
where ε is a coupling strength. For τ=1, when εR→∞, the phases are contracted to a single point exp(2iΘ) on the unit circle. For small values of εR, we can expand (Equation 2) to the linear order and obtain the more familiar form of mean-field coupled circle maps with phase doubling
(10)$$\varphi_{n}\left(t+1\right)=2\varphi_{n}\left(t\right)+\varepsilon R\left(t\right)\sin\left(2\Theta \left(t\right)-2\varphi_{n}\left(t\right)\right)+O\left(\varepsilon^{2}R^{2}\left(t\right)\right).$$

The crucial observation is that on the Ott-Antonsen manifold, the mean-field Z=Rexp(iΘ) transforms exactly the same way via (Equation 2), (Equation 3) as each element z=exp(iφ) on the unit circle [12,13]; that is,
(11)$$Z\left(t+1\right)=M(Z^{2}\left(t\right),\Phi \left(t\right),\tau \left(t\right)).$$

It is highly unusual that a closed analytic expression for the dynamics of the mean field can be derived and thus analyzed in coupled nonlinear dynamical systems. The reduction in infinitely dimensional microscopic dynamics to the low-dimensional dynamics of the mean-field [16] has been tremendously successful in the analysis of synchronization phenomena over the last decade, while the effects of the finite system size *N* remain difficult to analyze [19,20]. We note that the point measure of a finite ensemble of phases is never actually on the Ott-Antonsen manifold, but can, in some sense, be arbitrarily close to the so-called thermodynamic limit, i.e., the limit of the infinite system size N→∞.

Applying the Ott-Antonsen ansatz to networks of phase oscillators is possible if the network structure allows for the partitioning of the vertices into a few classes of equivalent vertices. Assuming that all vertices of a class are subjected to the same sinusoidal forcing, the dynamics of the phases in the network can be reduced to the dynamics of coupled mean fields on the Ott-Antonsen manifold for each vertex class [10,21,22,23,24]. Additionally, heterogeneity in the oscillators and fluctuations in the forces can be incorporated into the mean field dynamics if they follow Cauchy distributions [25,26,27].

### 2.1. Mean-Field Analysis

The mean-field dynamics (Equation 11) can be written in terms of the polar representation
(12)$$\Theta \left(t+1\right)=f\left(\Theta \left(t\right)\right)\;\;\;\;\;\;\mathrm{and}\;\;\;\;\;R\left(t+1\right)=\frac{\tau \left(t\right)+R^{2}\left(t\right)}{1+\tau \left(t\right)R^{2}\left(t\right)}.$$

This means that the dynamics of the phase Θ decouples from the amplitude and will evolve chaotically. Using Equations (Equation 9) and (Equation 12), we obtain the amplitude dynamics
(13)$$R\left(t+1\right)=\frac{\tanh\left(\frac{1}{2}\varepsilon R\left(t\right)\right)+R^{2}\left(t\right)}{1+\tanh\left(\frac{1}{2}\varepsilon R\left(t\right)\right)R^{2}\left(t\right)}$$
which describes the exact evolution of the order parameter *R* in a closed form. We can readily determine the fixed points of the mean-field amplitude R(t) and their linear stability. Both the complete synchronization R=1 and the complete desynchronization R=0 are fixed points of (Equation 13), and change stability at unique critical points $\varepsilon_{1}=\ln\left(2\right)\approx 0.69$ and $\varepsilon_{0}=2$, respectively, as determined by the eigenvalues of Jacobian of Equation (Equation 13) at these fixed points. These critical points are connected by an unstable fixed point branch $\left(\varepsilon \left(R_{u}\right),R_{u}\right)$, where
(14)$$\varepsilon \left(R_{u}\right)=\frac{1}{R_{u}}\log\left(\frac{{\left(1+R_{u}\right)}^{2}}{1+R_{u}^{2}}\right).$$

This expression is derived from (Equation 13) by setting $R\left(t+1\right)=R\left(t\right)=R_{u}$ and resolving the equation for ε.

This means that this system of all-to-all coupled, identical chaotic phase maps will always evolve to complete synchronization or complete desynchronization, with a small region ln(2)<ε<2 of bistability (Figure 2a).

### 2.2. Extension to Networks

Next, we have studied the same phase dynamics on a random network of *N* maps which are coupled to exactly *k* different, random neighbors. Here, each phase $\varphi_{n}$ couples to a local mean field
(15)$$Q_{n}=R_{n}e^{i\Theta_{n}}=\frac{1}{k}\sum_{n=1}^{N}A_{nm}z_{m}$$
where $A_{nm}$ are the entries of the adjacency matrix, i.e., equal to one if there is a link from vertex *m* to vertex *n*, but zero otherwise, and $k=\sum_{m=1}^{N}A_{nm}$ is the in-degree of node *n*, which, for computational simplicity, we assume to be identical for all nodes. Thus, with $\tau_{n}=\tanh\left(\frac{\varepsilon }{2}R_{n}\right)$, the dynamics of the phases coupled through a network are
(16)$$z_{n}\left(t+1\right)=\frac{e^{2i\Theta_{n}\left(t\right)}\tau_{n}\left(t\right)+z_{n}^{2}\left(t\right)}{1+e^{-2i\Theta_{n}\left(t\right)}\tau_{n}\left(t\right)z_{n}^{2}\left(t\right)}.$$

A class of networks is dense if $\lim_{N\to \infty }\langle k\rangle /N=p>0$, where 〈k〉 is the mean node degree. Therefore, *p* is the fraction of nodes, in relation to the system size *N*, that an oscillator is coupled to. Since dense networks are defined in the limit of N→∞, there is no sharp distinction between sparse and dense networks of finite size. We refer to a finite network as dense if two nodes share more than one neighbor on average, i.e., ${\langle k\rangle }^{2}/N=p^{2}N>1$. In large dense networks, the local mean fields of the oscillators in the neighborhood of each node (Equation 15) are equal to the global mean field, with a deviation of $O(1/\sqrt{k})$, where *k* is the size of the neighborhood, i.e., the in-degree of the node. Therefore, mean-field theory should be exact for dense networks in the thermodynamic limit where 〈k〉→∞.

*The network model*. First, we wish to compare the simulation results directly with our mean-field analysis. For large random networks with a link density p=k/N and 0<p<1, the numerical simulations are time-consuming since the *N* local mean fields at each node in the network need to be computed in each time step. To simplify these computations, we use a random network where each node couples to exactly *k* different random neighbors. This model with a unique in-degree of *k* for each node is slightly different from the Erdös Renyi model, with a Poissonian in-degree distribution of small relative width $\mathrm{std}\left(k\right)/\langle k\rangle ∼1/\sqrt{k}$. For large *k*, the results of the simulations in our random network model and other random networks with uncorrelated node degrees and a vanishing relative width of the degree distribution are expected to be identical.

## 3. Results

### 3.1. Distributions of Transient Times

We perform a large number *M* of simulations m=1…M from independent, uniformly distributed random initial phases over a maximum of *T* steps and record in each simulation the first time step $t_{m}$ when R≥0.5, i.e., the transition time from an incoherent state to complete synchronization. Finite-size scaling for such a discontinuous transition is challenging [28]. The exponential distribution of the times $t_{m}$, according to some characteristic transition rate, can be checked in a rank plot of time points $t_{m}$, which gives the sample complementary cumulative distribution $C\left(t\right)=\mathrm{prob}\left(t\geq t_{m}\right)=\mathrm{rank}\left(t_{m}\right)/M$ (Figure 3a,d).

An exponential tail distribution C(t) up to observation time *T* indicates an exponential distribution of transient times. Since the simulation time is finite, transition times $t_{m}\geq T$ are not observed, which represents a problem when we are interested in the average time to synchronization. However, assuming a discrete exponential, i.e., geometric distribution, a maximum likelihood estimation of the average transition time is possible up to values considerably exceeding the observation time *T* (see Appendix A).

Denoting the number of simulations that synchronize at times $t_{m}<T$ as $M_{T}$, and defining the observable values $l_{m}=\min\left(t_{m},T\right)$, the maximum likelihood estimation of the expected value $T_{esc}=E\left[t_{m}\right]$ for the geometric distribution is
(17)$$T_{esc}=\frac{\langle l_{m}\rangle M}{M_{T}}.$$
with the sample mean $\langle l_{m}\rangle$. If the transition to synchronization is observed in all simulations, i.e., $M_{T}=M$, the estimator is simply the sample mean of $t_{m}$, which is an estimator of $T_{esc}$ for arbitrary transient time distributions. However, when most runs do not synchronize within the finite simulation time *T*, the ratio $M/M_{T}$ contains additional information, and the estimated mean escape time can be much larger than the observation time.

### 3.2. During Coexistence: Escape over the Unstable Branch

In [29], it was reported that the transition from incoherence to collective dynamics in sparse networks of coupled logistic maps is of the mean-field type. The analysis in [30] predicts a shift in the critical coupling strength in random networks of Kuramoto phase oscillators of the order ${\langle k\rangle }^{2}/\langle k^{2}\rangle$ due to degree inhomogeneity, and 1/〈k〉 due to finite size fluctuations of the local mean fields. That is, in dense, homogeneous networks with ${\langle k\rangle }^{2}/\langle k^{2}\rangle \to 1$ and 〈k〉→∞, the critical coupling strength does not change. We expected to find similar behaviors for network-coupled Bernoulli maps. In complete or almost complete networks k/N=p≈1 for ε<2, there is a small probability that finite size fluctuations bring the order parameter *R* above the unstable branch, leading to a spontaneous transition to complete synchronization, as shown in Figure 4a. We first observe the scaling of the transient time in fully connected networks with p=1. For values of $\varepsilon <\varepsilon_{0}=2.0$, the transition rate to synchronization scales strongly with the size *N* of the system (Figure 3b,c). However, for values $\varepsilon >\varepsilon_{0}$, the average transition time depends very weakly on *N*, as the system grows exponentially fast from a state of incoherence, with $R\approx 1/\sqrt{N}$. We estimate a finite size scaling exponent β below the transition threshold by collapsing the curves $T_{esc}\left(\varepsilon ,N\right)$ using the ansatz $T_{esc}\left(\varepsilon ,N\right)=T_{esc}\left(\left(\varepsilon -\varepsilon_{0}\right)N^{\beta }\right).$ The data are consistent with an ad hoc exponent of β=1/3 (Figure 3c).

### 3.3. Above the Critical Coupling Strength: Long Chaotic Transient

Above the critical coupling strength $\varepsilon >\varepsilon_{0}=2$, we expected finite size fluctuations to grow exponentially fast and independently of *N*, as predicted by linear stability analysis of the mean-field Equation (Equation 13). Instead, for small connection probabilities 0<p<1, we have observed a chaotic transient with seemingly stationary finite size fluctuations $O(1/\sqrt{N})$ of the mean field (Figure 4). In the large *N* limit, the distribution of the transient times depends on the link density *p* with increasingly long transients as *p* is decreased, but it is otherwise independent of *N*.

A coupling strength for which a transition to complete synchronization could still be observed within the simulation time was considerably larger than the mean-field critical coupling $\varepsilon_{0}=2$. That is, even in dense networks and above the mean-field critical coupling, finite size fluctuations will not necessarily result in the nucleation and exponential growth of a collective mode. Such a delayed transition to synchronization [31] has so far not been described in systems of coupled phase oscillators [30,32,33] or coupled logistic maps [29].

In Figure 3f, we plot $T_{esc}$ over $(\varepsilon -\varepsilon_{0})p$ to demonstrate that the average transition time is roughly scaling as 1/p. We do not look for higher-order corrections such as a weak dependence of $\varepsilon_{0}$ on *p*, although the curves do not collapse perfectly. Note that the escape time is largely independent of the network size (Figure 3e,f). For p=0.1, 0.05, and 0.025 we have performed simulations with $N=10^{4}$ (circles) and with $N=5\times 10^{4}$ (crosses) for comparison. For p=0.01, we compare network sizes $N=10^{4}$ (circles) with very time-consuming simulations in networks with $N=10^{5}$ (crosses).

### 3.4. Discussion of Finite Size Scaling

Mean field theory assumes a phase distribution on the Ott-Antonsen manifold. The characteristic function of a wrapped Cauchy distribution is the geometric sequence $Z_{q}=Z^{q}$ of circular moments (Equation 6). However, in the incoherent state with *N* independent uniformly distributed phases $\varphi_{n}$ the circular moments of an ensemble
(18)$$Z_{q}=\frac{1}{N}\sum_{n=1}^{N}e^{iq\varphi_{n}}$$
are almost independent complex numbers with a Gaussian distribution of mean zero and a variance of 1/N by virtue of the central limit theorem. The action of the Bernoulli map on the circular moments is the shift
(19)$$Z_{q}\to Z_{2q},$$
that is, it is achieved by discarding all odd circular moments. The exponential growth of the order parameter in accordance to mean field theory is expected after the distribution comes close to the Ott-Antonsen manifold, i.e., when the first few circular moments align by chance sufficiently under the mapping (Equation 19); in particular, $Z_{2}\left(t\right)\approx Z_{1}^{2}\left(t\right)$. Unless the directions of $Z_{2}$ and $Z_{1}^{2}$ align by chance, as they would on the Ott-Antonsen manifold, the subsequent contraction of strength εR in the direction of $Z_{1}^{2}$ after the phase doubling may even decrease the amplitude of the order parameter. In addition, for coupling strengths ε below the critical value, $R=|Z_{1}|$ must be above the unstable branch $R>R_{u}\left(\varepsilon \right)∼\left(\varepsilon_{0}-\varepsilon \right)$.

The rate of such a random event should depend on the ratio between $R_{u}\left(\varepsilon \right)$ and the standard deviation $1/\sqrt{N}$ of the Gaussian distribution of the complex mean field. Based on this scaling argument, the expected time to synchronize should scale as $T_{esc}=T_{esc}\left(\left(\varepsilon -\varepsilon_{0}\right)\sqrt{N}\right)$ below the critical coupling. The best collapse of the estimated escape times in fully connected networks of coupled Bernoulli maps was observed by scaling the distance to $\varepsilon_{0}$ with $N^{1/3}$ (Figure 4c), i.e., the exponential divergence of the escape time approaches $\varepsilon_{0}$ slower than $1/\sqrt{N}$ in the thermodynamic limit. One possibility for this discrepancy is that the scaling argument only considers the chance of $R>R_{u}$ and not the alignment process of the higher-order circular moments.

Above the critical coupling strength, there is only the condition of the alignment of circular moments with the Ott-Antonsen manifold for the initiation of exponential growth. Since in the incoherent state, all circular moments are random Gaussian with identical variance, the alignment process (Equation 19) is strictly independent of the system size *N*. Once exponential growth in the direction of the Ott-Antonsen manifold occurs, the time to synchronization is logarithmic, that is, it is weakly dependent on *N*. However, it appears that the alignment with the Ott-Antonsen manifold needs to be stronger for networks with link densities of p<1. For small link densities, the divergence of the escape time occurs at larger values $\varepsilon >\varepsilon_{0}$. This is reminiscent of stabilization by noise [34], where a system is driven away from a low-dimensional unstable manifold of a fixed point into stronger attracting stable directions.

In simulations of dense random networks of coupled Bernoulli maps, we could see the independence of the mean escape time from the network size and the scaling of the escape time with roughly ∼1/p (Figure 3f). To explain this scaling, we argue that mean field theory might be extended to dense networks, where each node couples to a finite neighborhood of pN nodes in the network, and for every two nodes, these neighborhoods overlap on a set of size $p^{2}N$ (Figure 2b). The local mean fields are Gaussian random forces of mean value *Z*, variance 1/k=1/pN, and a pairwise correlation of *p*, which is the relative size of the overlap. The decrease in correlation between the local mean fields in networks with link densities p<1 can be interpreted as individual, finite size noise on the maps, which couple to the global mean field, plus some uncorrelated random deviation. Therefore, the contractions of the phases do not occur in the same direction for different nodes in the network. The strength of the contraction in the direction of the mean field is effectively reduced by the factor *p*, i.e.,
(20)$$\begin{array}{c} \tau =\tanh\left(\frac{1}{2}\varepsilon R\right)p\approx \frac{1}{2}\varepsilon pR \end{array}$$
shifting the coupling strength dependence of the transition time (above $\varepsilon_{0}$) by a factor of 1/p.

## 4. Conclusions

We have investigated the synchronization of coupled chaotic maps in dense random networks, utilizing mean-field equations and examining network configurations with different link probabilities. Firstly, we noticed the existence of chaotic transients to synchronization within these networks. This means that the incoherent state can persist for extended periods before transitioning into synchronization. This finding led us to study the statistics of transient times and their scaling behaviors in the process of synchronization. The transition times follow exponential distributions, indicating spontaneous transitions at a constant rate. It is noteworthy that the transition from incoherence to complete synchronization only occurs spontaneously in networks of finite size. Additionally, we have observed a remarkable dependence of the transient times to synchronization on the link probability *p*, represented by the ratio of the in-degree to the total number of nodes, at coupling strengths where an immediate transition to synchrony would be expected from mean field theory. Whether such a delayed transition is due to the specifics of our model or is typical for a more general class of dynamics remains an open question.

## Figures and Tables

**Figure 1 entropy-25-00983-f001:**
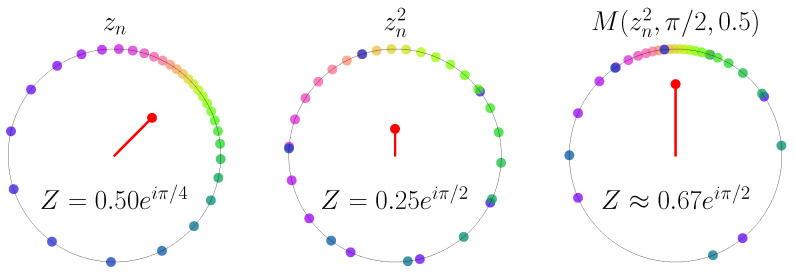
**Dynamics of phases**. N=30 points on the complex unit circle colored by phase, and corresponding mean field (red dot inside the unit circle). From left to right: initial phase configuration at the points $z_{n}$ with mean-field amplitude R=0.5 and mean-field phase Θ=π/4, after chaotic phase doubling $z_{n}^{2}$ with $R^{2}=0.25$ and 2Θ=π/2, and after subsequent contraction toward the angle π/2 with intensity τ=0.5.

**Figure 2 entropy-25-00983-f002:**
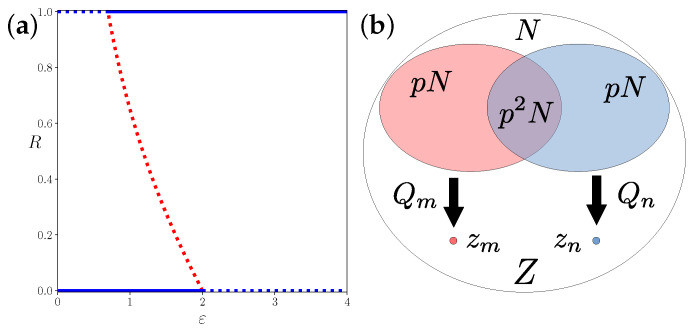
**Bifurcation diagram of the mean-field amplitude and a representation of the network interaction**. In (**a**), the bifurcation diagram of the all-to-all coupling mean-field dynamics (Equation 12), i.e., on the Ott-Antonsen manifold. Dotted lines show linearly unstable fixed points and solid lines show linearly stable fixed points in the thermodynamic limit. (**b**) Venn diagram of a dense network with *N* vertices and connection probability *p*. The sets of neighbors of nodes *m* and *n* are of size pN and their overlap is of size $p^{2}N$, resulting in correlated local mean fields $Q_{m}=R_{m}\exp\left(i\Theta_{m}\right)$ and $Q_{n}=R_{n}\exp\left(i\Theta_{n}\right)$ acting on the states $z_{m}$ and $z_{n}$. The ratio of the amplitudes of the local mean fields and the global mean field are independent of the network size *N*.

**Figure 3 entropy-25-00983-f003:**
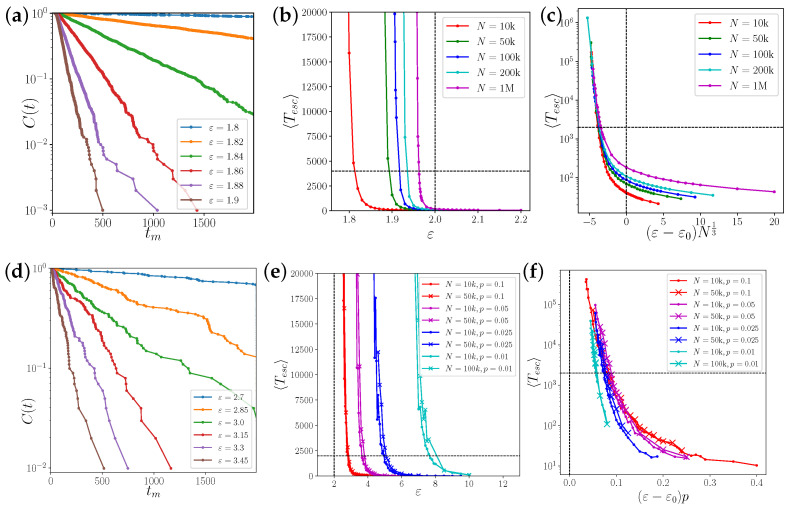
**Statistics of transient times $t_{m}$ to synchronization**. (**a**–**c**) In the fully connected network; (**d**–**f**) in random networks of various link densities p=k/N. The left panels show straight lines in semi-logarithmic plots of cumulative tail distributions of the transient times, demonstrating the rate character of the transition process. The middle panels show the estimated average transient times for various combinations of system sizes *N*, coupling strengths ε, and link densities *p*. The mean field critical coupling strength $\varepsilon_{0}=2.0$ and the maximum observation time *T* are marked by dashed lines. In the globally coupled system in pannels (**a**–**c**), the transient time depends strongly on the system size *N*, whereas in dense networks and above $\varepsilon_{0}$ (**d**–**f**), the transient time depends strongly on the link density p=k/N, but not on the system size. We demonstrate the scaling of the transient times in panels (**c**) and (**f**) on the right. In the globally coupled system, the exponential divergence of the transient times below $\varepsilon_{0}$ appears to be a function of $\left(\varepsilon -\varepsilon_{0}\right)N^{\frac{1}{3}}$. In dense networks, the exponential divergence is roughly a function of $(\varepsilon -\varepsilon_{0})p$.

**Figure 4 entropy-25-00983-f004:**
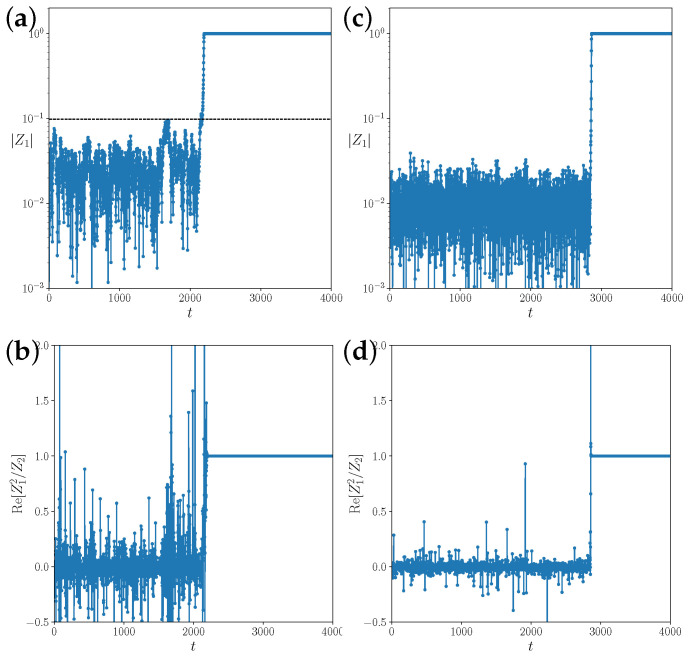
**Transient to synchronization** for N= 10,000 coupled maps in (**a**,**b**), a fully connected network with coupling strength ε=1.81 below the critical coupling $\varepsilon_{0}=2$, and (**c**,**d**), in a random network with connection probability p=0.1 for a coupling strength of ε=2.3 above the critical coupling. The upper panels (**a**,**c**) show the order parameter R(t), and the lower panels (**b**,**d**), the real part of the ratio of the first two circular moments Re[Z12/Z2]. This serves as a visual measure of the alignment of the system state with the Ott-Antonsen manifold, where the ratio is exactly equal to one. The dashed line in (**a**) marks the value of the unstable fixed point of the mean-field dynamics, $R_{u}=0.098$. Above that value, the state of complete synchronization is attractive on the Ott-Antonsen manifold. In (**b**,**d**), the incoherent state R=0 is unstable; however, finite size fluctuations do not grow exponentially. Instead, we observe a long chaotic transient.

## Data Availability

The Python code for the numerical simulations is available upon reasonable request from the corresponding author.

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
