# Peer review of "Exponentially Long Transient Time to Synchronization of Coupled Chaotic Circle Maps in Dense Random Networks"

_entropy, 2023, doi:10.3390/e25070983_

Round 1

Reviewer 1 Report

Dear Authors,

Your work is interesting and reveals novelty in the subject. However, the results deserve a more rigorous analysis. A case study could significantly improve your work. See the attachment. 

Best regards,

R

English should be reviewed.

Author Response

As requested by both reviewers, we have implemented a major revision of the manuscript. We have written a broader introduction, and restructured the sections to guide the reader better through our results. We have also included one additional figure visualizing the phase map dynamics. In our replies to the reviewers we try to answer the raised concerns and questions point by point. We hope that the reviewers will find the resubmitted manuscript in this improved form acceptable. 

Reviewer 2 Report

see attached file

looks overall ok

Author Response

(The authors gave the same response as above.)

Round 2

Reviewer 1 Report

Dear Authors,

The article benefited significantly from the corrections and changes made according to the reviewers' comments and suggestions. In my opinion, the article should be published in its current form.

Best regards,

The English language should be carefully reviewed.

Author Response

We thank the reviewer for their quick second round of reviewing of our manuscript.

Reviewer 2 Report

I thank the authors for their constructive reply to my previous report. They have taken my comments into account by resubmitting a profoundly rewritten ms. This looks like a proper scientific article now. I had no time to read in detail through the new ms., as MDPI only gave me 3 days, including the weekend. So all what I can say is that by surfacially looking at the revised ms., it seems ok now. Only two remarks: Seems there is a typo in Eq.(6), the integration variable is missing. And the authors may wish to review their conclusions, which have not been changed. I still find them unusually short. There would be more scope to deliver a proper message here.

Author Response

We thank the reviewer for their swift second reading of our manuscript and the positive feedback to our effort. In this last re-submission we have somewhat extended the conclusions section, as requested by the reviewer.